# Rural-urban differentials in the prevalence of diarrhoea among older adults in India: Evidence from Longitudinal Ageing Study in India, 2017–18

**Shobhit Srivastava**[1], **Snigdha Banerjee**[2], **Solomon Debbarma**[3], **Pradeep Kumar**[1]*, **Debashree Sinha**[3]

1 Department of Survey Research & Data Analytics, International Institute for Population Sciences, Mumbai, India, 2 Department of Family & Generations, International Institute for Population Sciences, Mumbai, India, 3 Department of Population & Development, International Institute for Population Sciences, Mumbai, India

* pradeepiips@yahoo.com

## Abstract

### Introduction

Diarrhoeal diseases are common among children and older adults. Yet, majority of the scientific studies deal with children, neglecting the other vulnerable and growing proportion of the population–the older adults. Therefore, the present study aims to find rural-urban differentials in the prevalence of diarrhoea among older adults in India and its states. Additionally, the study aims to find the correlates of diarrhoea among older adults in India. The study hypothesizes that there are no differences in the prevalence of diarrhoea in rural and urban areas.

### Methods

Data for this study was utilized from the recent Longitudinal Ageing Study in India (2017–18). The present study included eligible respondents aged 60 years and above (N = 31,464). Descriptive statistics along with bivariate analysis was presented to reveal the preliminary results. In addition, binary logistic regression analysis was used to fulfil the study objectives.

### Results

About 15% of older adults reported that they suffered from diarrhoea in the last two years. The prevalence of diarrhoea among older adults was found to be highest in Mizoram (33.5 per cent), followed by Chhattisgarh (30.7 per cent) and Bihar (30.2 per cent). There were significant rural-urban differences in the prevalence of diarrhoea among older adults in India (difference: 7.7 per cent). The highest rural-urban differences in the prevalence of diarrhoea were observed among older adults who were 80+ years old (difference: 13.6 per cent), used unimproved toilet facilities (difference: 12.7 per cent), lived in the kutcha house (difference: 10.2 per cent), and those who used unclean source of cooking fuel (difference: 9 per cent).

Institute for Population Sciences, Mumbai Institutional Data Access / Ethics Committee (contact via iipslasi@gmail.com; lasi@iips.net) for researchers who meet the criteria for access to confidential data.

**Funding:** The author(s) received no specific funding for this work.

**Competing interests:** The authors have declared that no competing interests exist.

Multivariate results show that the likelihood of diarrhoea was 17 per cent more among older adults who were 80+ years compared to those who belonged to 60–69 years' age group [AOR: 1.17; CI: 1.04–1.32]. Similarly, the older female had higher odds of diarrhoea than their male counterparts [AOR: 1.19; CI: 1.09–1.30]. The risk of diarrhoea had declined with the increase in the educational level of older adults. The likelihood of diarrhoea was significantly 32 per cent more among older adults who used unimproved toilet facilities than those who used improved toilet facilities [AOR: 1.32; CI: 1.21–1.45]. Similarly, older adults who used unimproved drinking water sources had higher odds of diarrhoea than their counterparts [AOR: 1.45; CI: 1.25–1.69]. Moreover, older adults who belonged to urban areas were 22 per cent less likely to suffer from diarrhoea compared to those who belonged to rural areas [AOR: 0.88; CI: 0.80–0.96].

## Conclusion

The findings of this study reveal that diarrhoea is a major health problem among older adults in India. There is an immediate need to address this public health concern by raising awareness about poor sanitation and unhygienic practices. With the support of the findings of the present study, policy makers can design interventions for reducing the massive burden of diarrhoea among older adults in rural India.

## Introduction

Diarrhoea is the second leading cause of mortality and morbidity throughout the world [2]. Although diarrhoeal diseases are common among children and older adults, death due to diarrhoea is three times more among older adults and specifically among those who belong in the population above 70 years of age than children under five years of age [1]. It not only causes physical discomfort but emotional distress as well. For instance, a study found out that older adults infected with diarrhoea experienced emotional distress since they had no control over faeces—when and where it would occur. Additionally, they lived in constant fear of experiencing faeces incontinence in public while they were away from home [2].

Diarrhoea among older adults is mostly caused due to an infection called 'shigella', that causes 18.4 deaths per lakh population [1]. Along with it unhygienic eating habits, contaminated food and water account for the continuing high prevalence of acute diarrhoea among older adults [3]. Infection can occur due to spoilt food, untreated water or from individual to individual [4]. It is also caused by a variety of bacterial, viral and parasitic organisms [5–8]. However, a study reveals that sometimes the causes of diarrhoea are not known [9] but it usually starts after two to four days after the infection and may last for three to seven days [10].

Current guidelines for the management of diarrhoea by the Ministry of Health and Family Welfare, Government of India, recommend a salt solution and zinc supplementation as precautionary steps that can prevent diarrhoea among older adults [11]. According to traditional medicine conventional ORS treatment with plant extracts can result in the reduction in the length of diarrhoeal symptoms [12]. A previous study based on a systematic review at the global level found that hand washing reduces diarrhoea by 40 per cent, but the practice of handwashing after contact with excreta is low throughout the world [13]. So as evidence suggest, this disease can easily be prevented by following very simple steps of hand washing, practicising safe drinking water, healthy hygiene and better sanitation [14].

Developing countries observe more cases of diarrhoea due to lack of safe drinking water, sanitation, and hygiene combined with poor nutritional status [15]. For example, in India, although negligible rural-urban difference is found in hand washing, almost 80 per cent of households in urban areas use soap and water to wash their hands compared to a maegre 49.4 per cent households in rural areas. Again, only 48.4 per cent of households have improved sanitation facilities, and 89.9 per cent have improved sources of drinking water. However, when improved sanitation facilities is bifurcated with place of residence, it is observed that 54 per cent of households in rural areas have no toilet facility compared to only 11 per cent households in the urban areas [16]. This rural-urban disparity in basic entitlements which is also the cause for illness due to diarrhoea encouraged us to take up the present study.

India's population over 60 years and above is projected to increase from 8 per cent in 2015 to 19 per cent in 2050 [17]. At the same time, 65 years and above population will increase from 6.4 per cent in 2019 to 8.6 per cent in 2030 [18]. Majority of the previous studies have focused on determinants of diarrhoea among children under five years of age in India [24–28], neglecting a vast and fast growing older adult population. On the other hand, acute diarrhoea is the most common diagnosis among older adults [19, 20]. Diarrhoea in developing countries like India, where there is poor sanitation and overcrowding [12, 21] is a major public health concern. Moreover, despite many governmental and non-governmental initiatives to restrict open defecation, Indians residing in rural areas still practise it, which is a cause for diarrhoeal infection [20–23].

Therefore, the present study is rationalised on the following arguments. One, based on the fact that the proportion of Indian older adults is increasing at an increasing rate and is likely to rise in the coming decades [17]. Two, considering that the older adults are at a high risk of being infected by diarrhoea and die due to diarrhoea. Three, research evidence suggests that people living in rural areas are more succeptible to diarrhoea because of poor sanitation. Finally, given the dearth of scientific studies on the prevalence of diarrhoea among older adults and its determinants in India, the present study aims to find the rural-urban differential in the prevalence of diarrhoea among older adults in India and its states. Additionally, the study aims to find the determinants of diarrhoea among older adults in India. The study hypothesize that there are no difference in the prevalence of diarrhoea among older adults in rural and urban areas.

## Methods

### Data

Data for this study was utilized from the Longitudinal Ageing Study in India (LASI) wave 1 [22]. LASI is a full-scale national survey of scientific investigation of India's health, economic, and social determinants and consequences of population ageing, conducted in 2017–18 [22]. LASI is a nationally representative survey of over 72000 older adults aged 45 and above across all states and India's union territories. The survey's main objective is to study the health status and the social and economic well-being of older adults in India. LASI adopted a multistage stratified area probability cluster sampling design to arrive at the eventual units of observation: older adults age 45 and above and their spouses irrespective of their age. The survey adopted a three-stage sampling design in rural areas and a four-stage sampling design in urban areas. In each state/UT, the first stage involved the selection of Primary Sampling Units (PSUs), that is, sub-districts (Tehsils/Talukas), and the second stage involved the selection of villages in rural areas and wards in urban areas in the selected PSUs. In rural areas, households were selected from selected villages in the third stage. However, sampling in urban areas involved an additional stage. Specifically, in the third stage, one Census Enumeration Block (CEB) was

randomly selected in each urban area. In the fourth stage, households were selected from this CEB. The detailed methodology was published in the survey report with the complete information on the survey design and data collection [22]. The present study included the eligible respondent's aged 60 years and above. The present study's total sample size was 31,464 (Rural-20,725 and Urban-10,739) older adults aged 60 years and above.

## Variable description

**Outcome variable.** The outcome variable was in binary form, i.e., diarrhoea (no and yes). The information was assessed by asking that "whether, in the past two years, the respondent was diagnosed with diarrhoea by a health professional?" The response was stated as no and yes [23, 24].

**Explanatory variables.** The main explanatory variable was a place of residence and it was coded as rural and urban area. The classification was defined as in previous literature. It was found that disease prevalence varies significantly by place of residence [25–29].

Age was coded as 60–69 years, 70–79 years and 80 and above; Sex was coded as male and female; Education was coded as no education/primary not completed, primary completed, secondary completed and higher and above; Marital status was coded as currently married, widowed and others which includes separated/divorced/never married; Working status was coded as currently working, retired/not currently working and never worked; Overweight/obesity was coded as underweight, normal and overweight/obese. The respondents having a body mass index of 25 and above were categorized as obese/overweight.

Source of cooking fuel was coded as unclean and clean; Type of toilet facility was coded as unimproved and improved; Source of drinking water was coded as unimproved and improved, and type of house was coded as pucca, semi pucca and kutcha. The monthly per capita expenditure (MPCE) quintile was assessed using household consumption data. Sets of 11 and 29 questions on the expenditures on food and non-food items, respectively, were used to canvas the sample households. Food expenditure was collected based on a reference period of seven days, and non-food expenditure was collected based on reference periods of 30 days and 365 days. Food and non-food expenditures have been standardized to the 30-day reference period. The monthly per capita consumption expenditure (MPCE) is computed and used as the summary measure of consumption [22]. The variable was then divided into five quintiles, i.e., from poorest to richest. Religion was coded as Hindu, Muslim, Christian, and Others. Caste was coded as Scheduled Tribe, Scheduled Caste, Other Backward Class, and others. The Scheduled Caste includes a group of socially segregated population and by their financially/economically status as per the Hindu caste hierarchy. The Scheduled Castes (SCs) and Scheduled Tribes (STs) are among the India's most disadvantaged socio-economic groups. The OBC is the group of people who were identified as "educationally, economically and socially backward". The OBC's are considered low in the traditional caste hierarchy. The "other" caste category is identified as having higher social status [30–32]. Geographical region was coded as North, Central, East, Northeast, West, and South.

## Statistical analysis

Descriptive statistics and bivariate analysis were presented in the present study to reveal the preliminary results. Proportion test [33] was used to find the significance level for residential differences for diarrhoea prevalence. Moreover, binary logistic regression analysis [34] was used to analyse the association between the outcome variable (diarrhoea) and other explanatory variables.

The binary logistic regression model is usually put into a more compact form as follows:

$$\text{Logit}[P(Y = 1)] = \beta_0 + \beta * X + \epsilon$$

The parameter $\beta_0$ estimates the log odds of diarrhoea for the reference group, while $\beta$ estimates the maximum likelihood, the differential log odds of diarrhoea associated with a set of predictors X, as compared to the reference group, and $\epsilon$ represents the residual in the model. The variance inflation factor (VIF) was used to check for the existence of multicollinearity, and the test found that there was no confirmation of multicollinearity [35, 36].

## Results

### Socio-demographic profile of study population (Table 1)

About 58 per cent of older adults belonged to the 60–69 years' age cohort, 30 per cent were in the age group of 70–79, and the rest of (11 per cent) older adults belonged to the 80+ years, age

**Table 1. Socio-demographic and economic profile of older adults in India, 2017–18.**

| Background characteristics | Rural | | Urban | | Total | |
|---|---|---|---|---|---|---|
| | Sample | % | Sample | % | Sample | % |
| **Age (in years)** | | | | | | |
| 60–69 | 12139 | 58.6 | 6268 | 58.4 | 18410 | 58.5 |
| 70–79 | 6169 | 29.8 | 3354 | 31.2 | 9501 | 30.2 |
| 80+ | 2417 | 11.7 | 1117 | 10.4 | 3553 | 11.3 |
| **Sex** | | | | | | |
| Male | 10045 | 48.5 | 4835 | 45.0 | 14931 | 47.5 |
| Female | 10680 | 51.5 | 5904 | 55.0 | 16533 | 52.6 |
| **Education** | | | | | | |
| No education/primary not completed | 15984 | 77.1 | 4937 | 46.0 | 21381 | 68.0 |
| Primary completed | 2069 | 10.0 | 1511 | 14.1 | 3520 | 11.2 |
| Secondary completed | 1988 | 9.6 | 2598 | 24.2 | 4371 | 13.9 |
| Higher and above | 682 | 3.3 | 1693 | 15.8 | 2191 | 7.0 |
| **Marital status** | | | | | | |
| Currently married | 13017 | 62.8 | 6315 | 58.8 | 19391 | 61.6 |
| Widowed | 7280 | 35.1 | 4162 | 38.8 | 11389 | 36.2 |
| Others | 427 | 2.1 | 262 | 2.4 | 684 | 2.2 |
| **Body Mass Index** | | | | | | |
| Underweight | 6062 | 32.4 | 1142 | 12.2 | 7406 | 23.5 |
| Normal | 9742 | 52.1 | 4561 | 48.7 | 14203 | 45.1 |
| Overweight/obese | 2884 | 15.4 | 3658 | 39.1 | 6153 | 19.6 |
| **Working status** | | | | | | |
| Currently working | 7341 | 35.4 | 2106 | 19.6 | 9680 | 30.8 |
| Retired/currently not working | 8774 | 42.3 | 4719 | 43.9 | 13470 | 42.8 |
| Never worked | 4610 | 22.2 | 3913 | 36.4 | 8314 | 26.4 |
| **MPCE quintile** | | | | | | |
| Poorest | 4446 | 21.5 | 2396 | 22.3 | 6829 | 21.7 |
| Poorer | 4608 | 22.2 | 2197 | 20.5 | 6831 | 21.7 |
| Middle | 4375 | 21.1 | 2207 | 20.6 | 6590 | 21.0 |
| Richer | 3932 | 19.0 | 2117 | 19.7 | 6038 | 19.2 |
| Richest | 3364 | 16.2 | 1822 | 17.0 | 5175 | 16.5 |
| **Religion** | | | | | | |

*(Continued)*

**Table 1.** (Continued)

| Background characteristics | Rural | | Urban | | Total | |
|---|---|---|---|---|---|---|
| | Sample | % | Sample | % | Sample | % |
| Hindu | 17309 | 83.5 | 8497 | 79.1 | 25871 | 82.2 |
| Muslim | 2021 | 9.8 | 1604 | 14.9 | 3548 | 11.3 |
| Christian | 623 | 3.0 | 269 | 2.5 | 900 | 2.9 |
| Others | 772 | 3.7 | 369 | 3.4 | 1145 | 3.6 |
| **Caste** | | | | | | |
| Scheduled Caste | 4572 | 22.1 | 1220 | 11.4 | 5949 | 18.9 |
| Scheduled Tribe | 2125 | 10.3 | 325 | 3.0 | 2556 | 8.1 |
| Other Backward Class | 9213 | 44.5 | 5056 | 47.1 | 14231 | 45.2 |
| Others | 4815 | 23.2 | 4139 | 38.5 | 8729 | 27.7 |
| **Place of residence** | | | | | | |
| Rural | | | | | 22196 | 70.6 |
| Urban | | | | | 9268 | 29.5 |
| **Source of cooking fuel** | | | | | | |
| Unclean | 13455 | 64.9 | 1984 | 18.5 | 16122 | 51.2 |
| Clean | 7270 | 35.1 | 8755 | 81.5 | 15342 | 48.8 |
| **Type of toilet facility** | | | | | | |
| Unimproved | 8035 | 38.8 | 1319 | 12.3 | 9744 | 31.0 |
| Improved | 12690 | 61.2 | 9420 | 87.7 | 21720 | 69.0 |
| **Source of drinking water** | | | | | | |
| Unimproved | 1200 | 5.8 | 1594 | 14.8 | 2660 | 8.5 |
| Improved | 19525 | 94.2 | 9145 | 85.2 | 28804 | 91.5 |
| **Type of House** | | | | | | |
| Pucca | 8512 | 41.8 | 8281 | 80.0 | 16015 | 50.9 |
| Semi pucca | 7064 | 34.7 | 1646 | 15.9 | 9931 | 31.6 |
| Kutcha | 4794 | 23.5 | 428 | 4.1 | 5519 | 17.5 |
| **Region** | | | | | | |
| North | 2655 | 12.8 | 1293 | 12.0 | 3960 | 12.6 |
| Central | 4920 | 23.7 | 1533 | 14.3 | 6593 | 21.0 |
| East | 5678 | 27.4 | 1573 | 14.7 | 7439 | 23.6 |
| Northeast | 691 | 3.3 | 226 | 2.1 | 935 | 3.0 |
| West | 2898 | 14.0 | 2662 | 24.8 | 5401 | 17.2 |
| South | 3883 | 18.7 | 3451 | 32.1 | 7136 | 22.7 |
| **Total** | 20,725 | 100.0 | 10,739 | 100.0 | 31464 | 100.0 |

group. A higher proportion of older adults from rural areas had no education/primary not completed (77 per cent), whereas, in urban areas, about 46 per cent of older adults had no education. About one-third and 12 per cent of older adults from rural and urban areas were underweight. Nearly 35 per cent and 20 per cent of older adults were currently working in rural and urban areas, respectively. Around 35 per cent of older adults in rural areas used clean cooking fuel, which was more than double in urban areas (81.5 per cent). In rural areas, three-fifth of older adults used improved toilet facilities while in urban areas, 88 per cent of older adults used improved toilet facilities. Moreover, a higher proportion of older adults from rural and urban areas used improved drinking water sources. About 42 per cent of older adults in rural areas lived in the pucca house, and this proportion was almost double in urban areas than in rural counterparts.

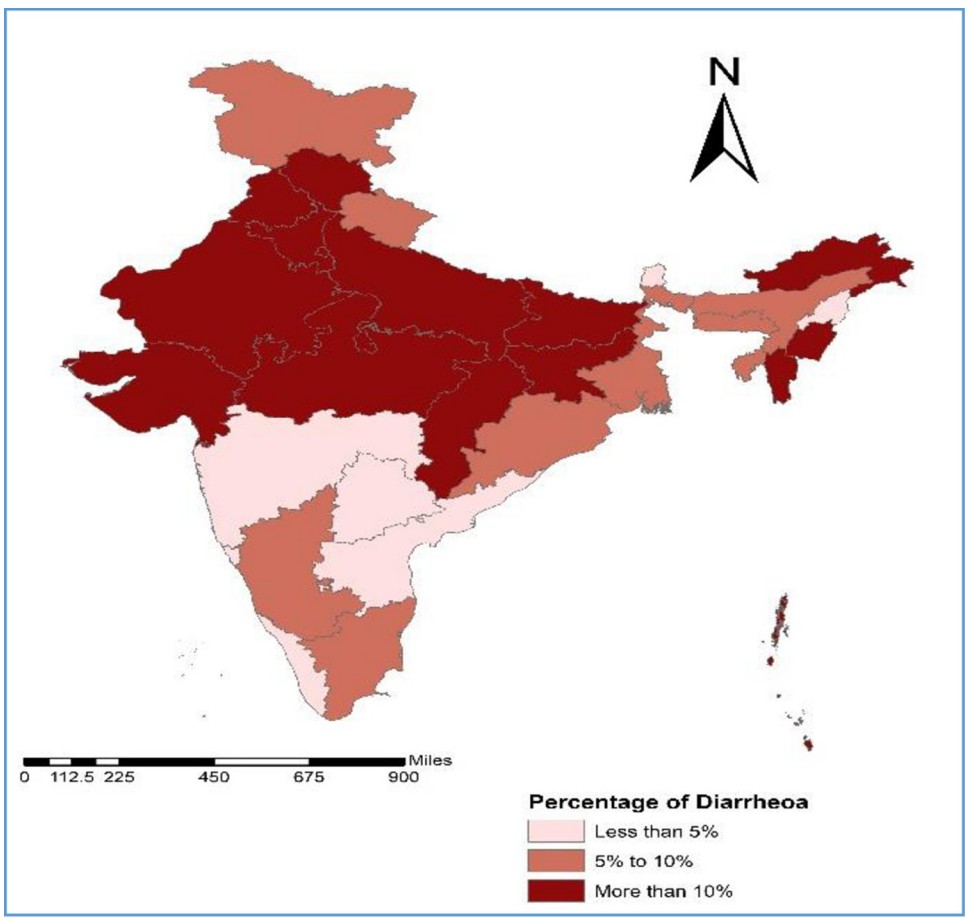

**Fig 1. Prevalence of diarrhoea among older adults by states of India, 2017–18.**

**Fig 1** displays the prevalence of diarrhoea among older adults in the states of India. About 15 per cent of older adults in India suffer from diarrhoea (rural-17 per cent and urban-9 per cent). The prevalence of diarrhoea among older adults was highest in Mizoram (33.5 per cent), followed by Chhattisgarh (30.7 per cent), Bihar (30.2 per cent), and Rajasthan (30.2 per cent). Moreover, in rural areas, this prevalence was highest in Mizoram (33.2 per cent), followed by Chhattisgarh (32.6 per cent), Rajasthan (32.2 per cent) and Bihar (30 per cent) (Table 2). In the case of urban India, the highest prevalence of diarrhoea among older adults was observed in Mizoram (34 per cent), followed by Bihar (32.1 per cent), Haryana (25.9 per cent), Himachal Pradesh (25.7 per cent), and Madhya Pradesh (24.5 per cent) (Table 2).

## Rural-urban differential in the prevalence of diarrhoea among older adults in India (Table 3)

Overall, the result shows a significant rural-urban difference in the prevalence of diarrhoea among older adults in India (difference: 7.7 per cent). The prevalence of diarrhoea was significantly higher among 80+ years older adults (17.6 per cent) than other age group. It has a negative association with the educational level of older adults. For instance, the prevalence of diarrhoea decreased with the increase in the level of education among older adults. A similar pattern was observed in rural as well urban areas. Diarrhoea was more prevalent among underweight older adults, and it was also true for rural and urban areas. Wealth quintile had negative

**Table 2. Percentage of older adults suffered from diarrhoea in states of India, 2017–18.**

| States | Rural (%) | Urban (%) | Total (%) |
|---|---|---|---|
| Jammu & Kashmir | 8.9 | 4.0 | 7.2 |
| Himachal Pradesh | 19.9 | 25.7 | 20.1 |
| Punjab | 9.7 | 15.9 | 11.1 |
| Chandigarh | 0.0 | 8.7 | 8.4 |
| Uttarakhand | 7.3 | 9.7 | 7.8 |
| Haryana | 24.1 | 25.9 | 24.5 |
| Delhi | 0.0 | 12.9 | 12.9 |
| Rajasthan | 32.2 | 23.0 | 30.2 |
| Uttar Pradesh | 27.9 | 19.9 | 26.4 |
| Bihar | 30.0 | 32.1 | 30.2 |
| Arunachal Pradesh | 16.1 | 8.2 | 15.6 |
| Nagaland | 0.1 | 0.0 | 0.1 |
| Manipur | 18.1 | 22.2 | 20.3 |
| Mizoram | 33.2 | 34.0 | 33.5 |
| Tripura | 5.8 | 5.8 | 5.8 |
| Meghalaya | 6.2 | 5.5 | 6.1 |
| Assam | 7.3 | 2.6 | 6.5 |
| West Bengal | 8.9 | 5.0 | 7.9 |
| Jharkhand | 11.9 | 8.3 | 11.2 |
| Odisha | 6.3 | 5.1 | 6.2 |
| Chhattisgarh | 32.6 | 23.1 | 30.7 |
| Madhya Pradesh | 30.0 | 24.5 | 28.8 |
| Gujarat | 17.7 | 12.6 | 15.1 |
| Daman & Diu | 12.3 | 6.3 | 8.3 |
| Dadra & Nagar Haveli | 23.1 | 20.6 | 22.1 |
| Maharashtra | 5.7 | 1.8 | 4.2 |
| Andhra Pradesh | 2.0 | 0.0 | 1.5 |
| Karnataka | 11.3 | 1.8 | 6.5 |
| Goa | 4.4 | 1.6 | 2.9 |
| Lakshadweep | 2.4 | 1.2 | 1.6 |
| Kerala | 3.0 | 3.3 | 3.3 |
| Tamil Nadu | 5.6 | 4.0 | 5.1 |
| Puducherry | 5.3 | 0.4 | 2.4 |
| Andaman & Nicobar Island | 20.2 | 16.0 | 19.8 |
| Telangana | 0.9 | 0.9 | 0.9 |
| **India** | **17.1** | **9.4** | **14.8** |

association with the prevalence of diarrhoea, moreover it was higher in rural areas in all wealth groups than urban areas. The prevalence of diarrhoea was higher among older adults who used unclean cooking fuel (18.3 per cent) and those who used unimproved toilet facilities (20.3 per cent) compared to their counterparts. A similar result was observed for older adults who belonged to rural and urban areas. The highest rural-urban differences in the prevalence of diarrhoea were observed among older adults who were 80+ years old (difference: 13.6 per cent), used unimproved toilet facilities (difference: 12.7 per cent), lived in the kutcha house (difference: 10.2 per cent), and those who used unclean source of cooking fuel (difference: 9 per cent). Older adults who used improved drinking water (15.2%) reported more diarrhea

**Table 3. Percentage of older adults suffering from diarrhoea by their background characteristics in India, 2017–18.**

| Background characteristics | Total | Rural | Urban | Differences | p-value |
|---|---|---|---|---|---|
| | % | % | % | % | |
| **Age (in years)** | | | | | |
| 60–69 | 14.2 | 16.3 | 9.2 | 7.0 | <0.001 |
| 70–79 | 15.0 | 17.0 | 10.2 | 6.8 | <0.001 |
| 80+ | 17.6 | 21.3 | 7.7 | 13.6 | <0.001 |
| **Sex** | | | | | |
| Male | 14.8 | 16.9 | 9.1 | 7.7 | <0.001 |
| Female | 14.9 | 17.3 | 9.5 | 7.7 | <0.001 |
| **Education** | | | | | |
| No education/primary not completed | 16.3 | 17.8 | 10.5 | 7.3 | <0.001 |
| Primary completed | 14.0 | 16.2 | 10.3 | 5.9 | <0.001 |
| Secondary completed | 11.3 | 14.6 | 8.0 | 6.6 | <0.001 |
| Higher and above | 8.3 | 10.7 | 7.1 | 3.6 | <0.001 |
| **Marital status** | | | | | |
| Currently married | 14.5 | 16.5 | 9.4 | 7.1 | <0.001 |
| Widowed | 15.4 | 18.2 | 9.5 | 8.6 | <0.001 |
| Others | 13.1 | 16.3 | 6.7 | 9.7 | <0.001 |
| **Body Mass Index** | | | | | |
| Underweight | 19.1 | 20.1 | 12.3 | 7.8 | <0.001 |
| Normal | 14.9 | 16.4 | 10.8 | 5.6 | <0.001 |
| Overweight/obese | 10.9 | 14.9 | 7.0 | 7.9 | <0.001 |
| **Working status** | | | | | |
| Currently working | 15.6 | 16.8 | 10.2 | 6.6 | <0.001 |
| Retired/currently not working | 14.8 | 17.1 | 9.4 | 7.7 | <0.001 |
| Never worked | 13.9 | 17.4 | 8.8 | 8.5 | <0.001 |
| **MPCE quintile** | | | | | |
| Poorest | 15.8 | 18.2 | 10.4 | 7.8 | <0.001 |
| Poorer | 17.0 | 19.1 | 11.4 | 7.7 | <0.001 |
| Middle | 14.1 | 16.0 | 9.2 | 6.9 | <0.001 |
| Richer | 13.6 | 16.1 | 7.9 | 8.3 | <0.001 |
| Richest | 12.9 | 15.2 | 7.5 | 7.8 | <0.001 |
| **Religion** | | | | | |
| Hindu | 15.1 | 17.5 | 9.0 | 8.5 | <0.001 |
| Muslim | 16.3 | 18.7 | 12.2 | 6.6 | <0.001 |
| Christian | 7.3 | 7.6 | 6.3 | 1.3 | 0.349 |
| Others | 9.2 | 9.8 | 7.6 | 2.2 | 0.915 |
| **Caste** | | | | | |
| Scheduled Caste | 15.8 | 17.3 | 8.9 | 8.4 | <0.001 |
| Scheduled Tribe | 16.2 | 16.4 | 14.4 | 1.9 | 0.043 |
| Other Backward Class | 15.0 | 17.8 | 8.3 | 9.4 | <0.001 |
| Others | 13.6 | 15.8 | 10.3 | 5.5 | <0.001 |
| **Place of residence** | | | | | |
| Rural | 17.1 | | | | |
| Urban | 9.4 | | | | |
| **Source of cooking fuel** | | | | | |
| Unclean | 18.3 | 19.2 | 10.2 | 9.0 | <0.001 |
| Clean | 11.2 | 13.1 | 9.2 | 4.0 | <0.001 |

*(Continued)*

**Table 3.** (Continued)

| Background characteristics | Total | Rural | Urban | Differences | p-value |
|---|---|---|---|---|---|
| | % | % | % | % | |
| **Type of toilet facility** | | | | | |
| Unimproved | 20.3 | 21.6 | 8.9 | 12.7 | <0.001 |
| Improved | 12.4 | 14.2 | 9.4 | 4.8 | <0.001 |
| **Source of drinking water** | | | | | |
| Unimproved | 10.9 | 12.9 | 8.8 | 4.1 | 0.038 |
| Improved | 15.2 | 17.3 | 9.4 | 7.9 | <0.001 |
| **Type of house** | | | | | |
| Pucca | 12.6 | 15.0 | 9.3 | 5.8 | <0.001 |
| Semi pucca | 15.3 | 16.7 | 10.9 | 5.8 | <0.001 |
| Kutcha | 20.6 | 21.2 | 11.1 | 10.2 | 0.003 |
| **Region** | | | | | |
| North | 20.8 | 22.3 | 17.0 | 5.4 | <0.001 |
| Central | 27.5 | 28.9 | 21.9 | 7.1 | <0.001 |
| East | 16.4 | 17.5 | 11.8 | 5.7 | <0.001 |
| Northeast | 8.6 | 8.4 | 9.2 | -0.8 | <0.001 |
| West | 7.4 | 8.8 | 5.6 | 3.2 | <0.001 |
| South | 4.2 | 5.5 | 2.4 | 3.1 | <0.001 |
| **Total** | 14.8 | 17.1 | 9.4 | 7.7 | <0.001 |

Difference = Rural-Urban.

than those who used unimproved drinking water (10.9%). Underweight older adults had a higher prevalence of diarrhoea irrespective of their place of residence.

## Estimates from multivariate analysis for older adults who suffered from diarrhoea in India (Table 4)

The result depicts that the likelihood of diarrhoea was 17 per cent more likely among older adults who were 80+ years compared to those who belonged to the 60–69 years age group [AOR: 1.17; CI: 1.04–1.32]. Similarly, the older female had higher odds of diarrhoea than older male counterparts [AOR: 1.19; CI: 1.09–1.30]. Older adults with no education/primary not completed had higher odds to suffer from diarrhoea in reference to older adults with higher and above education [AOR: 1.43; CI:1.20,1.71]. With reference to scheduled caste older adults, scheduled tribe and other backward class older adults had 22 per cent and 24 per cent higher risk of diarrhoea, respectively. Older adults who belonged to urban areas were 22 per cent less likely to suffer from diarrhoea than those who belonged to rural areas [AOR: 0.88; CI: 0.80–0.96]. The risk of diarrhoea among older adults was higher in the Central region, whereas it was lower in other parts of India compared to the North region. The likelihood of diarrhoea was significantly 32 per cent more likely among older adults who used an unimproved toilet facilities than those who used improved toilet facilities [AOR: 1.32; CI: 1.21–1.45]. Similarly, older adults who used unimproved drinking water sources had higher odds of diarrhoea than their counterparts [AOR: 1.45; CI: 1.25–1.69].

## Discussion

Although diarrhoeal diseases are common in older populations [19, 37], there is a paucity of study on them, making the preventable disease a major cause of concern. The present study

**Table 4. Logistic regression estimates for older adults who suffered from diarrhoea by their background characteristics in India, 2017–18.**

| Background characteristics | AOR |
|---|---|
| | 95% CI |
| **Age (in years)** | |
| 60–69 | Ref. |
| 70–79 | 1.08(0.99,1.17) |
| 80+ | 1.17*(1.04,1.32) |
| **Sex** | |
| Male | Ref. |
| Female | 1.19*(1.09,1.30) |
| **Education** | |
| No education/primary not completed | 1.43*(1.20,1.71) |
| Primary completed | 1.33*(1.09,1.60) |
| Secondary completed | 1.31*(1.10,1.58) |
| Higher and above | Ref. |
| **Marital status** | |
| Currently married | Ref. |
| Widowed | 1.08(0.98,1.17) |
| Others | 1.01(0.80,1.28) |
| **Body Mass Index** | |
| Underweight | 1.02(0.91,1.15) |
| Normal | 1.07(0.97,1.18) |
| Overweight/obese | Ref. |
| **Working status** | |
| Currently working | Ref. |
| Retired/currently not working | 0.96(0.88,1.04) |
| Never worked | 0.80*(0.72,0.89) |
| **MPCE quintile** | |
| Poorest | 0.85*(0.75,0.96) |
| Poorer | 1.01(0.90,1.13) |
| Middle | 0.89*(0.79,1.02) |
| Richer | 0.97(0.87,1.09) |
| Richest | Ref. |
| **Religion** | |
| Hindu | Ref. |
| Muslim | 0.93(0.83,1.04) |
| Christian | 1.19*(1.01,1.41) |
| Others | 0.63*(0.53,0.76) |
| **Caste** | |
| Scheduled Caste | Ref. |
| Scheduled Tribe | 1.22*(1.07,1.39) |
| Other Backward Class | 1.24*(1.12,1.37) |
| Others | 0.96(0.86,1.07) |
| **Place of residence** | |
| Rural | Ref. |
| Urban | 0.88*(0.80,0.96) |
| **Source of cooking fuel** | |
| Unclean | 1.03(0.94,1.12) |

(*Continued*)

**Table 4.** (Continued)

| Background characteristics | AOR |
|---|---|
| | 95% CI |
| Clean | Ref. |
| **Type of toilet facility** | |
| Unimproved | 1.32*(1.21,1.45) |
| Improved | Ref. |
| **Source of drinking water** | |
| Unimproved | 1.45*(1.25,1.69) |
| Improved | Ref. |
| **Type of house** | |
| Pucca | Ref. |
| Semi pucca | 1.21*(1.11,1.32) |
| Kutcha | 1.07(0.97,1.19) |
| **Region** | |
| North | Ref. |
| Central | 1.43*(1.29,1.6) |
| East | 0.71*(0.64,0.79) |
| Northeast | 0.46*(0.39,0.54) |
| West | 0.38*(0.33,0.43) |
| South | 0.18*(0.15,0.2) |

Ref: Reference

* if p<0.05; CI: Confidence interval; AOR: Adjusted Odds Ratio.

analysed data from Longitudinal Ageing Study in India to estimate diarrhoeal prevalence among older adults in India and across its states. A significant rural-urban difference in the prevalence of diarrhoea among older adults is found. Those who are living in rural areas are more likely to suffer from the disease. Using unimproved drinking water, unimproved sanitation facility, and low access to health care facilities in rural areas are found to be positively associated with a high prevalence of diarrhoea [38, 39]. Furthermore, literary evidences mostly on childhood diarrhoea show that environmnetal as well as personal hygiene to be significant risk factors of acute diarrhoea among rural population [40, 41].

The study also found out a high prevalence of diarrhoea among underweight older adults who belonged to rural areas compared to urban areas. Improper nutrition among older adults who reside in rural areas could be a possible explanation for this finding as evidence from previous analysis on children showed undernutrition as an underlying cause associated with diarrhoea [42]. Again, a study on children in a rural community in South India showed that undernourished children had a higher risk for acute diarrhoea [40].

Drawing similarities from research on children in Indonesia, Bangladesh, Ethiopia [40–43] which emphasize that children who lived in houses with less dirty sewage, utilized latrine facilities, belonged to households where handwashing was practiced before preparing food had significantly lower diarrhoea prevalence, our study results exhibit that older adults who used unimproved toilet facility had higher odds of suffering from diarrhoea. Contradicting our result which shows that older adults with no education had higher likelihood of suffering from diarrhoea, a study on incidence and determinants of acute diarrhoea among Malaysian population showed that those with higher level of education had higher likelihood of acute diarrhoea [43].

Logistic regression results reveal that the prevalence of diarrhoea was positively associated with higher age of older adults, who belonged to Scheduled Tribe (22 per cent higher risk) and OBC social group (24 per cent higher risk). The finding is consistent with a study carried out among under-five children in India [44]. Moreover, the study reveals that older adults who belong to the Christian religion were more likely to have diarrhoeal risk than Hindu older adults. However, this finding is inconsistent with previous research on under-five children in India [38, 44].

Generally, the incidence of diarrhoea remains a tremendous burden on population from low- and middle-income countries due to multiple determinants such as low socioeconomic status, lack of safe drinking water, inadequate sanitation, poor hygiene and crowding but the present study contradicts the existing literature and shows that the odds of older adults suffering from diarrhoea is higher among those who belonged to a richer section of the population [38, 45]. Probable explanations for this finding could be: 1) A high prevalence of diabetes among older adults, in general and those belonging to high Socio Economic Status [46–48] and because diabetic diarrheoa is a major gastrointestinal discomfort [49, 50], older adults belonging to the richer section may have a high prevalence of diarrhoea. 2) Since multimorbidity is higher among older adults [51, 52], older adults may be consuming medicines that may cause diarrhoea.

Earlier studies on children under five in India, have shown regional disparity in the prevalence of diarrhoea [37, 48]. The present study shows a higher concentration of diarrhoea among older adults in central and northeastern parts of the country compared to the southern states of India [53]. The finding shows similarity with studies based on children in India [54]. This could be because of unequal access to health care facilities, use of untreated drinking water and low hygienic practices. The regional disparity in the prevalence of diarrhoea among older adults in India highlight the need for spatial studies to identify the hotspots that will help in the planning of controlling the disease.

## Strengths and limitations of the study

The study contributes to the growing body of research documenting the high prevalence of diarrhoea in India, especially in rural areas among older adults and highlights the disease's predictors. The primary strength of the study lies in the use of countrywide data on older adults. Earlier studies on diarrhoea focused on a particular region with smaller sample size and on children under five years of age [55, 56]. However, research evidences on diarrhoeal diseases among older adults is scarce [57]. The study has certain limitations too. First, diarrheoal prevalence was based on self-reporting and recall of the respondents; this leaves a scope for underreporting of diarrhoea's prevalence. Second, the study is based on one time point data, therefore trend could not be established. Third, the illustration of the causal relationship between diarrhoea and geriatric outcomes was also limited as we used a cross-sectional study design. Lastly, evidence suggests that hand wash plays a vital role in the incidence of diarrhoea. However, the absence of information on hand washing practice before preparation of food prevented us from examining its association with the incidence of diarrhoea among older adults.

## Conclusion

The study found a high prevalence of diarrhoea among older adults residing in rural areas. Since, diarrhoea is caused due to public health challenges posed by poor sanitation, unhygenic practices like unsafe drinking water and lack of hand washing, policies should be implemented in rural areas in terms of spreading awareness of sanitation and hygiene practices. Thus, the findings of this study can be used to design target interventions for reducing the massive

burden of diarrhoea among older adults in India. Also, as India is undergoing an epidemiological transition along with demographic transition, research on disease burden owing to acute diarrhoea and its associated risk factors among older adults need to be studied.

## Acknowledgments

Authors would like to acknowledge Ms Adrita Banerjee for helping in the editing of the manuscript.

## Author Contributions

**Conceptualization:** Shobhit Srivastava, Pradeep Kumar.

**Data curation:** Shobhit Srivastava.

**Formal analysis:** Shobhit Srivastava.

**Methodology:** Shobhit Srivastava, Pradeep Kumar.

**Software:** Shobhit Srivastava.

**Supervision:** Solomon Debbarma, Pradeep Kumar.

**Validation:** Snigdha Banerjee, Solomon Debbarma, Pradeep Kumar.

**Visualization:** Snigdha Banerjee, Pradeep Kumar, Debashree Sinha.

**Writing – original draft:** Snigdha Banerjee, Solomon Debbarma.

**Writing – review & editing:** Solomon Debbarma, Pradeep Kumar, Debashree Sinha.

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
