## [Decision Letter · Decision Letter 0]

29 Jun 2021

PONE-D-21-09569

Rural-urban differentials in prevalence of diarrhoea among older in India: An evidence from Longitudinal Ageing Study in India, 2017-18

PLOS ONE

Dear Dr. Kumar,

Thank you for submitting your manuscript to PLOS ONE. After careful consideration, we feel that it has merit but does not fully meet PLOS ONE’s publication criteria as it currently stands. Therefore, we invite you to submit a revised version of the manuscript that addresses the points raised during the review process.

We look forward to receiving your revised manuscript.

Kind regards,

Shah Md Atiqul Haq

Academic Editor

PLOS ONE

Additional Editor Comments:

Dear Authors,

I would like to ask you to revise the article based on the reviewers' comments and suggestions.

Please focus on the methodology and conclusion section.

Best wishes,

Journal Requirements:

2. PLOS ONE does not copy edit accepted manuscripts (https://journals.plos.org/plosone/s/criteria-for-publication#loc-5). To that effect, please ensure that your submission is free of typos and grammatical errors.

5. We note that Figure 1 in your submission contain map images which may be copyrighted. All PLOS content is published under the Creative Commons Attribution License (CC BY 4.0), which means that the manuscript, images, and Supporting Information files will be freely available online, and any third party is permitted to access, download, copy, distribute, and use these materials in any way, even commercially, with proper attribution. For these reasons, we cannot publish previously copyrighted maps or satellite images created using proprietary data, such as Google software (Google Maps, Street View, and Earth). For more information, see our copyright guidelines: http://journals.plos.org/plosone/s/licenses-and-copyright.

You may seek permission from the original copyright holder of Figure 1 to publish the content specifically under the CC BY 4.0 license. 

If you are unable to obtain permission from the original copyright holder to publish these figures under the CC BY 4.0 license or if the copyright holder’s requirements are incompatible with the CC BY 4.0 license, please either i) remove the figure or ii) supply a replacement figure that complies with the CC BY 4.0 license. Please check copyright information on all replacement figures and update the figure caption with source information. If applicable, please specify in the figure caption text when a figure is similar but not identical to the original image and is therefore for illustrative purposes only.

Reviewers' comments:

Reviewer's Responses to Questions

**Comments to the Author**

1. Is the manuscript technically sound, and do the data support the conclusions?

Reviewer #1: Partly

Reviewer #2: Yes

2. Has the statistical analysis been performed appropriately and rigorously? 

Reviewer #1: Yes

Reviewer #2: Yes

3. Have the authors made all data underlying the findings in their manuscript fully available?

Reviewer #1: Yes

Reviewer #2: Yes

4. Is the manuscript presented in an intelligible fashion and written in standard English?

Reviewer #1: No

Reviewer #2: Yes

5. Review Comments to the Author

Reviewer #1: Abstract: The abstract contains incomplete sentences and needs to be rewritten. No mention of rural urban differentials could be found in the objectives, which according to the title is supposedly the main aim of the paper. Sentence like "Descriptive statistics along with bivariate analysis was presented in the present study to reveal the preliminary analysis. " is not clear. Do the authors mean preliminary results?

In the abstract, Authors may first present the overall scenario in India and states and then can move to the rural urban differential and then the multivariate results.

The policy recommendations written in the abstract are not coming directly from the study. Authors should recommend policies or need based on their findings and results. This is a very broad recommendation. Authors should try to write the recommendations linking with their study results.

Authors should choose keywords more attentively. Using rural urban differential would be better than key words of prevalence and regression.

Introduction has many information but has to be reframed. There should be link and should be written with continuity and flow. Authors may only write Diarrhoea in place of Diarrhoea diseases . They can also write diarrhoeal disease but diarrhoea disease is not recommended.

"So present study focus on the older adults in India who are above 60 years of age and are suffering from diarrhoea." This sentence is very confusing and it seems that the authors only chose the older adults suffering from diarrhoea?

"Unlike children, the study found that diarrhoea is associated with emotional distress among the older adults" Which study? the authors should describe a little more while writing about any other study. In that way it will be easier to read.

" Acute diarrhoea leads to a substantial disease burden worldwide and most commonly diagnoses among the older adults [6, 7]. This is common in developing countries like India, where there are poor sanitation and overcrowding. Global Burden of Diseases, in 2016 estimated, diarrhoea was the eighth leading cause of mortality, responsible for more than 6 million deaths [8, 9]."--- India and global data are getting mixed up. Authors may first discuss about global and then on India specifically.

"A previous study found that hand washing reduces diarrhoea by 40 per cent, but the practice of handwashing after contact with excreta is low throughout the world "---Where was the study conducted and among whom? give more information.

Rural urban differential as mentioned in the title is missing from objectives.

"Data for this study was utilized from the recent release of the Longitudinal Ageing Study in India (LASI) wave 1 "-- sentences should be more simpler.

"The present study is conducted on the eligible respondent’s age 60 years and above. "--- or included?

Authors may simply describe about the variables and their categories. Giving references for each categories may be avoided as there are more than hundreds of research papers using the same variables and its categories. These are all established variables.

Results

Socio-demographic profile of study population (Table 1)---The study is on aged population, but the authors did not mention about the age categories here and percentage of elderly under each category.

Prevalence of diarrhoea among older adults in India (Table 2)--Author should first present about the overall scenario of states and India and then can focus on the rural urban differentials and other aspects.

There should be separate subheading and paragraph for rural urban differentials as this is one of the important aspect of the study.

Figure 1 displays the prevalence of diarrhoea .......... --- this should be written in a more presentable manner. India %? Give total column in Table 3

Estimates from multivariate analysis for older adults who suffered from diarrhoea in India (Table 4)--Consider rewriting and reframing some of the subheadings

Discussion-- needs to be rewritten. This discussion part is almost like Literature review. Authors may go through few literature and see how to frame the discussion part. The studies quoted in the discussion should support your findings (or contrary) and should not be written separately. Should be linked to your study findings. For example "A previous study based on rural Bangladesh suggested that hand washing before preparing food is particularly important to prevent diarrhoea [55, 56]. " With which finding from the present study are the authors linking this study in support or in contrary. There are many literatures mentioned like this is the discussion without linking them to the study results.

There are few portions under discussion which will be more appropriate for the need /scope of the study part.

"In the context of the increasingly ageing trend in India, the prevalence and correlates of agents among older diarrheal patients was needed to explore"-- not clear

"The research shows a significant rural-urban difference in the prevalence of diarrhoea among older adults in India"--- Should write whether it is high in rural or urban too

How are the results considering religion and economic condition?

Citing references should be done properly and only where necessary.

"Research related to the prevalence of diarrhoea among the geriatric age group should also be emphasized as the issue is growing at an unprecedented pace globally"-- which issue? Issue of ageing or diarrhoea? Recommended to write more clearly.

Separate section on strengths and limitations can be written other than merging with the discussion part .

Need to rewrite conclusion part. Focus on the main contribution from the paper. Try focusing on policy recommendations coming directly from the study.

Few references needs to be modified according to referencing style.

Tables- total column may be given in Table 1, 2, 3.

Table 2- In results section the urban rural and total percentage by few important background characteristics may be explained as a background before going to the rural urban differential.

Table 4- May consider reordering of the variables. First may give soci0 demographic, then economic and household variables.

I congratulate the authors for selecting this topic and working extensively on the literature review and analysis. But they have to revise the manuscript as the result, discussion, conclusion parts needs to be rewritten. The main findings from this study are getting disoriented and lost. They should also focus on the conclusion and policy recommendation part as this is a very important topic.

Reviewer #2: The manuscript sounds good, I recommened to accept the paper for publication. Although I have some observations. First is that author should rewrite the discussion part. As I found there is very less linking between the variables consisting older adults and diarrhoea among older. Also In discussion part author has quitely written the literature references to explain and support the current study results. But I think, he should consider the theme as a whole rather than going point by point. Second is that author should discuss more about the logistic regression and literature references to support his findings.

6. PLOS authors have the option to publish the peer review history of their article (what does this mean?). If published, this will include your full peer review and any attached files.

Reviewer #1: No

Reviewer #2: **Yes: **Tushar Dakua

---

## [Author Response · Author response to Decision Letter 0]

21 Jul 2021

Review Comments to the Author

Reviewer #1: Abstract: The abstract contains incomplete sentences and needs to be rewritten. No mention of rural urban differentials could be found in the objectives, which according to the title is supposedly the main aim of the paper. Sentence like "Descriptive statistics along with bivariate analysis was presented in the present study to reveal the preliminary analysis. " is not clear. Do the authors mean preliminary results?

Response: Dear reviewer, I agree with your comment. The abstract is now rewritten. Moreover, preliminary analysis is now written as preliminary results.

In the abstract, Authors may first present the overall scenario in India and states and then can move to the rural urban differential and then the multivariate results.

Response: Comment incorporated. 

The policy recommendations written in the abstract are not coming directly from the study. Authors should recommend policies or need based on their findings and results. This is a very broad recommendation. Authors should try to write the recommendations linking with their study results.

Response: The recommendation is now updated. 

Authors should choose keywords more attentively. Using rural urban differential would be better than key words of prevalence and regression.

Response: Thanks for the suggestion. Amendment has been done.

Introduction has many information but has to be reframed. There should be link and should be written with continuity and flow. Authors may only write Diarrhoea in place of Diarrhoea diseases. They can also write diarrhoeal disease but diarrhoea disease is not recommended.

Response: Thanks for the suggestion, change has been made in the manuscript. 

"So present study focus on the older adults in India who are above 60 years of age and are suffering from diarrhoea." This sentence is very confusing and it seems that the authors only chose the older adults suffering from diarrhoea?

Response: This sentence has been reframed, to clarify that this study include all population in this age group. 

"Unlike children, the study found that diarrhoea is associated with emotional distress among the older adults" Which study? the authors should describe a little more while writing about any other study. In that way it will be easier to read.

Response: Few additional line on study area and target population is added in the manuscript. 

"Acute diarrhoea leads to a substantial disease burden worldwide and most commonly diagnoses among the older adults [6, 7]. This is common in developing countries like India, where there are poor sanitation and overcrowding. Global Burden of Diseases, in 2016 estimated, diarrhoea was the eighth leading cause of mortality, responsible for more than 6 million deaths [8, 9]."--- India and global data are getting mixed up. Authors may first discuss about global and then on India specifically.

Response: Data has been put in sequence as suggested. 

"A previous study found that hand washing reduces diarrhoea by 40 per cent, but the practice of handwashing after contact with excreta is low throughout the world "---Where was the study conducted and among whom? give more information.

Response: The study population referred in this particular study is being mentioned as per the suggestion.

Rural urban differential as mentioned in the title is missing from objectives.

"Data for this study was utilized from the recent release of the Longitudinal Ageing Study in India (LASI) wave 1 "-- sentences should be more simpler.

Response: Rural-urban differential has been mentioned in the objective. Sentence has been modified.

"The present study is conducted on the eligible respondent’s age 60 years and above. "--- or included?

Response: Thanks for the pointing out. Modification has been done.

Authors may simply describe about the variables and their categories. Giving references for each categories may be avoided as there are more than hundreds of research papers using the same variables and its categories. These are all established variables.

Response: Comment incorporated. 

Results

Socio-demographic profile of study population (Table 1)---The study is on aged population, but the authors did not mention about the age categories here and percentage of elderly under each category.

Response: Percentage under each age categories has been mentioned.

Prevalence of diarrhoea among older adults in India (Table 2)--Author should first present about the overall scenario of states and India and then can focus on the rural urban differentials and other aspects.

Response: Comment incorporated.

There should be separate subheading and paragraph for rural urban differentials as this is one of the important aspect of the study.

Response: Comment incorporated.

Figure 1 displays the prevalence of diarrhoea .......... --- this should be written in a more presentable manner. India %? Give total column in Table 3

Response: Comment incorporated.

Estimates from multivariate analysis for older adults who suffered from diarrhoea in India (Table 4)--Consider rewriting and reframing some of the subheadings

Response: Amendment has been done.

Discussion-- needs to be rewritten. This discussion part is almost like Literature review. Authors may go through few literature and see how to frame the discussion part. The studies quoted in the discussion should support your findings (or contrary) and should not be written separately. Should be linked to your study findings. For example "A previous study based on rural Bangladesh suggested that hand washing before preparing food is particularly important to prevent diarrhoea [55, 56]. " With which finding from the present study are the authors linking this study in support or in contrary. There are many literatures mentioned like this is the discussion without linking them to the study results.

Response: Amendment has been done.

There are few portions under discussion which will be more appropriate for the need /scope of the study part.

"In the context of the increasingly ageing trend in India, the prevalence and correlates of agents among older diarrheal patients was needed to explore"-- not clear

Response: The discussion section has been revised accordingly

"The research shows a significant rural-urban difference in the prevalence of diarrhoea among older adults in India"--- Should write whether it is high in rural or urban too

Response: Comment incorporated

How are the results considering religion and economic condition?

Response: Thank you for pointing out this comment. These are discussed in discussion section. 

Citing references should be done properly and only where necessary.

Response: Amendment has been done.

"Research related to the prevalence of diarrhoea among the geriatric age group should also be emphasized as the issue is growing at an unprecedented pace globally"-- which issue? Issue of ageing or diarrhoea? Recommended to write more clearly.

Response: Amendment has been done.

Separate section on strengths and limitations can be written other than merging with the discussion part.

Response: Separate section on strengths and limitations has been done.

Need to rewrite conclusion part. Focus on the main contribution from the paper. Try focusing on policy recommendations coming directly from the study.

Response: Amendment has been done

Few references needs to be modified according to referencing style.

Response: Comment incorporated. 

Tables- total column may be given in Table 1, 2, 3.

Response: Total column has been added in Table 1, 2, and 3.

Table 2- In results section the urban rural and total percentage by few important background characteristics may be explained as a background before going to the rural urban differential.

Response: Changes have been made as per the suggestion.

Table 4- May consider reordering of the variables. First may give soci0 demographic, then economic and household variables.

Response: Changes have been made as per the suggestion.

I congratulate the authors for selecting this topic and working extensively on the literature review and analysis. But they have to revise the manuscript as the result, discussion, conclusion parts needs to be rewritten. The main findings from this study are getting disoriented and lost. They should also focus on the conclusion and policy recommendation part as this is a very important topic.

Response:

Reviewer #2: The manuscript sounds good, I recommened to accept the paper for publication. Although I have some observations. First is that author should rewrite the discussion part. As I found there is very less linking between the variables consisting older adults and diarrhoea among older. Also In discussion part author has quitely written the literature references to explain and support the current study results. But I think, he should consider the theme as a whole rather than going point by point. Second is that author should discuss more about the logistic regression and literature references to support his findings.

Response: Revised the discussion accordingly.

---

## [Decision Letter · Decision Letter 1]

19 Nov 2021

PONE-D-21-09569R1Rural-urban differentials in prevalence of diarrhoea among older in India: An evidence from Longitudinal Ageing Study in India, 2017-18PLOS ONE

Dear Dr. Kumar,

Thank you for submitting your manuscript to PLOS ONE. After careful consideration, we feel that it has merit but does not fully meet PLOS ONE’s publication criteria as it currently stands. Therefore, we invite you to submit a revised version of the manuscript that addresses the points raised during the review process. Please submit your revised manuscript by 10 weeks. If you will need more time than this to complete your revisions, please reply to this message or contact the journal office at plosone@plos.org. Please include the following items when submitting your revised manuscript:A rebuttal letter that responds to each point raised by the academic editor and reviewer(s). You should upload this letter as a separate file labeled 'Response to Reviewers'.A marked-up copy of your manuscript that highlights changes made to the original version. You should upload this as a separate file labeled 'Revised Manuscript with Track Changes'.An unmarked version of your revised paper without tracked changes. You should upload this as a separate file labeled 'Manuscript'.

We look forward to receiving your revised manuscript.

Kind regards,

Shah Md Atiqul Haq

Academic Editor

PLOS ONE

Journal Requirements:

Additional Editor Comments (if provided):

Dear authors,

I would like to ask you to read the reviewers' comments and suggestions carefully.

The reviewers still find so many shortcomings in the paper.

I suggest to revise the paper and resubmit it. The revised version could be sent to new reviewers.

Best wishes,

Reviewers' comments:

Reviewer's Responses to Questions

**Comments to the Author**

1. If the authors have adequately addressed your comments raised in a previous round of review and you feel that this manuscript is now acceptable for publication, you may indicate that here to bypass the “Comments to the Author” section, enter your conflict of interest statement in the “Confidential to Editor” section, and submit your "Accept" recommendation.

Reviewer #1: (No Response)

Reviewer #2: All comments have been addressed

2. Is the manuscript technically sound, and do the data support the conclusions?

Reviewer #1: (No Response)

Reviewer #2: Yes

3. Has the statistical analysis been performed appropriately and rigorously? 

Reviewer #1: (No Response)

Reviewer #2: Yes

4. Have the authors made all data underlying the findings in their manuscript fully available?

Reviewer #1: (No Response)

Reviewer #2: Yes

5. Is the manuscript presented in an intelligible fashion and written in standard English?

Reviewer #1: No

Reviewer #2: Yes

6. Review Comments to the Author

Reviewer #1: Although the authors have done few changes in the manuscript, most of the parts still need serious modifications. Authors should consider the following comments as constructive that will help them to make the manuscript more suitable for publication.

Sentences in the abstract has not been reframed. Authors are recommended to read all the sentences. English should be checked properly as there are noticeable problems with prepositions, verbs. Sudden use of words like “Moreover”, “While” and “however” throughout the paper should be avoided.

Sentences like “Diarrhoeal diseases are seen among all age group” is not at all recommended.

Authors have not rewritten the abstract as recommended in the last review comments.

Under Methods section “during 2016-2017” should be moved to any other sentence as it is not having any meaning in the present sentence. It seems authors have only taken into consideration the specific comments to be changed in the abstract. Policy recommendations are still very broad in the abstract part.

“About 15 per cent of older adults in India were suffered from diarrhoea” reframe with correct forms of verbs.

Discussion:

“This study shows that the prevalence of diarrhoea is 7.7 percentage points higher in rural areas than urban areas”- avoid repeating the percentage from the results in the discussion part. But authors should discuss about the results and the main crux.

“However, the finding is not similar with previous research in India [44, 51].” Should be reframed.

“Our study contradicts the existing literature and shows that the odds of older adults suffering from diarrhoea were higher among those who belonged to richer section of population [44, 53].

” Try to discuss this more as this is an important finding. Also check the analysis as the results are opposite in table 3.

Paragraph starting with “ Geographical differences in prevalence of diarrhoea” should be rewritten.

There should be more discussion on the important findings from the study. Also, the discussion ends abruptly.

Strengths and limitations

“Therefore, very few studies have dealt with the older adults [57].” Not clear

This section should be written more clearly as the authors are suddenly starting to write about the limitations. Everything is getting mixed up.

English needs to be checked throughout the manuscript. Errors in verbs, singular plural, prepositions can be found throughout the paper.

References: References were not checked according to the last comment provided. Many references are not upto date. Check references 12, 10 and try to update them if necessary.

Kumar Panda Leuven SK, Kumar Bastia A. Anti-diarrheal activities of medicinal plants of Similipal Biosphere Re-serve, Potential Antibacterial Agent(s) against Foodborne Pathogens View project. Int J Med Aromat Plants. – wrong reference.

Many references are not having publication year and page numbers.

There is no uniform style. In spite of giving comments in the first review about the references, the authors failed to check the references. Authors need to check each reference and write them properly. Avoid writing responses to comments as “changes incorporated” when authors have not done any changes in the reference section.

Tables

“May consider reordering of the variables. First may give soci0 demographic, then economic and household variables”

This comment was given previously also. But the authors did not do the reordering of variables in all the tables. But in the response, they have written that they have made the changes. They should start with age, sex and then the other socio demographic, economic and household variables. Though the authors have stated this specific comment has been incorporated, the same is not the case. They have not modified the variables.

In table 3

Source of drinking water - adults suffering from diarrhoea

Unimproved 10.9 %

Improved 15.2 %

This result is very shocking. Nothing has been mentioned about this result in the whole paper. On the other hand, the results are opposite in the table with logistics regression. Authors may check the analysis for both the tables. And then mention them in results and discussion.

Another important finding that “prevalence of diarrhoea was more among underweight older adults” has also not mentioned in the discussion section. Discussion section should be properly written with focus on the interesting findings from the study along with linking with the previous literature. Similarly, the issue of living in kutcha pakka house is also missing from the discussion. Authors should go through the tables, results section and then write the discussion.

Reviewer #2: The authors have worked so nice. The paper sounds good. I recommend editor to ask the author for some minor rivisions like:

1. Write the abstract in a comprehensive way. Not copy and paste from the manuscript.

2. Outcome variable is something which really comes out from the analysis. Not from the data set. So, authors can rename the outcome variable or can constract or recode the outcome variable.

7. PLOS authors have the option to publish the peer review history of their article (what does this mean?). If published, this will include your full peer review and any attached files.

Reviewer #1: No

Reviewer #2: **Yes: **Tushar Dakua

---

## [Author Response · Author response to Decision Letter 1]

7 Dec 2021

Dear authors,

I would like to ask you to read the reviewers' comments and suggestions carefully. The reviewers still find so many shortcomings in the paper. I suggest to revise the paper and resubmit it. The revised version could be sent to new reviewers.

Reviewer #1: Although the authors have done few changes in the manuscript, most of the parts still need serious modifications. Authors should consider the following comments as constructive that will help them to make the manuscript more suitable for publication. Sentences in the abstract has not been reframed. Authors are recommended to read all the sentences. English should be checked properly as there are noticeable problems with prepositions, verbs. Sudden use of words like “Moreover”, “While” and “however” throughout the paper should be avoided.

Response: The authors agree with the comment. The changes are now incorporated in the manuscript. 

Sentences like “Diarrhoeal diseases are seen among all age group” is not at all recommended.

Response: The sentence is now reframed. 

Authors have not rewritten the abstract as recommended in the last review comments.

Response: Dear reviewer, the entire abstract is now changed. 

Under Methods section “during 2016-2017” should be moved to any other sentence as it is not having any meaning in the present sentence. It seems authors have only taken into consideration the specific comments to be changed in the abstract. Policy recommendations are still very broad in the abstract part.

Response: The sentence is now reframed. Dear reviewer, the study found that diarrhoea among older adults is significantly more prevalent in rural areas than in urban areas. Therefore, authors recommend the policy makers to focus more on rural areas to reduce the overall residential gap specifically and in general to reduce the overall burden of diarrhoea among older adults. 

“About 15 per cent of older adults in India were suffered from diarrhoea” reframe with correct forms of verbs.

Response: The sentence is now reframed with correct form of verbs. 

Discussion:

“This study shows that the prevalence of diarrhoea is 7.7 percentage points higher in rural areas than urban areas”- avoid repeating the percentage from the results in the discussion part. But authors should discuss about the results and the main crux.

Response: The sentence is reframed. 

“However, the finding is not similar with previous research in India [44, 51].” Should be reframed.

Response: We have reframed the sentence.

“Our study contradicts the existing literature and shows that the odds of older adults suffering from diarrhoea were higher among those who belonged to richer section of population [44, 53].

” Try to discuss this more as this is an important finding. Also check the analysis as the results are opposite in table 3.

Response: Thank you for the comment. Given the dearth of literature that could possibly explain this unusual relationship, we will take up a deeper analysis of the economic status of the older adults and their odds of having diarrhoea in future. 

Paragraph starting with “Geographical differences in prevalence of diarrhoea” should be rewritten. There should be more discussion on the important findings from the study. Also, the discussion ends abruptly.

Response: Thank you for the comment. We have re-written the paragraph.

Strengths and limitations

“Therefore, very few studies have dealt with the older adults [57].” Not clear

This section should be written more clearly as the authors are suddenly starting to write about the limitations. Everything is getting mixed up.

Response: The authors agree with the comment. The sentence is now removed. The strength and limitation section is reframed for better understanding. 

English needs to be checked throughout the manuscript. Errors in verbs, singular plural, prepositions can be found throughout the paper.

Response: The paper is now edited by a native English speaker. 

References: References were not checked according to the last comment provided. Many references are not up to date. Check references 12, 10 and try to update them if necessary.

Kumar Panda Leuven SK, Kumar Bastia A. Anti-diarrheal activities of medicinal plants of Similipal Biosphere Re-serve, Potential Antibacterial Agent(s) against Foodborne Pathogens View project. Int J Med Aromat Plants. – wrong reference.

Response: References are changed

Many references are not having publication year and page numbers. There is no uniform style. In spite of giving comments in the first review about the references, the authors failed to check the references. Authors need to check each reference and write them properly. Avoid writing responses to comments as “changes incorporated” when authors have not done any changes in the reference section.

Response: References has now been changed

Tables

“May consider reordering of the variables. First may give soci0 demographic, then economic and household variables” This comment was given previously also. But the authors did not do the reordering of variables in all the tables. But in the response, they have written that they have made the changes. They should start with age, sex and then the other socio demographic, economic and household variables. Though the authors have stated this specific comment has been incorporated, the same is not the case. They have not modified the variables.

Response: Dear reviewer, authors had arranged the table 4 as per your suggestion. Apologize for not making change in table 1 & 3. Now we have arranged the table 1 & 3 too as per your suggestion. Kindly refer to Table 1, 3 & 4.

In table 3

Source of drinking water - adults suffering from diarrhea

Unimproved 10.9 %

Improved 15.2 %

This result is very shocking. Nothing has been mentioned about this result in the whole paper. On the other hand, the results are opposite in the table with logistics regression. Authors may check the analysis for both the tables. And then mention them in results and discussion.

Response: Dear reviewer, authors are also shocked by this inconsistency. We have checked the analysis again and found the same results. The reason for the opposite results in the logistic table might be because of the adjusted results. 

Discussion person can discuss this issue in discussion section. Another important finding that “prevalence of diarrhoea was more among underweight older adults” has also not mentioned in the discussion section. Discussion section should be properly written with focus on the interesting findings from the study along with linking with the previous literature. Similarly, the issue of living in kutcha pakka house is also missing from the discussion. Authors should go through the tables, results section and then write the discussion.

Response: We have tried to incorporate the important findings of the paper. However, due to dearth of literature on the topic, at times it is difficult to support it with literature. 

Reviewer #2: The authors have worked so nice. The paper sounds good. I recommend editor to ask the author for some minor revisions like:

1. Write the abstract in a comprehensive way. Not copy and paste from the manuscript.

Response: Dear reviewer, the authors edited the abstract as per your suggestion. 

2. Outcome variable is something which really comes out from the analysis. Not from the data set. So, authors can rename the outcome variable or can constract or recode the outcome variable.

Response: Dear reviewer, I agree with the comment. The outcome variable was assessed using the question “whether, in the past two years, the respondent was diagnosed with diarrhoea by a health professional?” The variable was coded as no and yes in the dataset.

---

## [Decision Letter · Decision Letter 2]

10 Jan 2022

PONE-D-21-09569R2Rural-urban differentials in the prevalence of diarrhoea among older adults in India: Evidence from Longitudinal Ageing Study in India, 2017-18PLOS ONE

Dear Dr. Kumar,

Thank you for submitting your manuscript to PLOS ONE. After careful consideration, we feel that it has merit but does not fully meet PLOS ONE’s publication criteria as it currently stands. Therefore, we invite you to submit a revised version of the manuscript that addresses the points raised during the review process.

We look forward to receiving your revised manuscript.

Kind regards,

Shah Md Atiqul Haq

Academic Editor

PLOS ONE

Additional Editor Comments (if provided):

Dear authors,

Please address the comments and suggestions of the reviewer.

One reviewer advises you to reject the essay with some valuable comments and suggestions.

If you are willing to address the reviewers' comments, please review them carefully.

Reviewers' comments:

Reviewer's Responses to Questions

**Comments to the Author**

1. If the authors have adequately addressed your comments raised in a previous round of review and you feel that this manuscript is now acceptable for publication, you may indicate that here to bypass the “Comments to the Author” section, enter your conflict of interest statement in the “Confidential to Editor” section, and submit your "Accept" recommendation.

Reviewer #1: (No Response)

Reviewer #2: All comments have been addressed

2. Is the manuscript technically sound, and do the data support the conclusions?

Reviewer #1: Partly

Reviewer #2: Yes

3. Has the statistical analysis been performed appropriately and rigorously? 

Reviewer #1: N/A

Reviewer #2: Yes

4. Have the authors made all data underlying the findings in their manuscript fully available?

Reviewer #1: Yes

Reviewer #2: Yes

5. Is the manuscript presented in an intelligible fashion and written in standard English?

Reviewer #1: Yes

Reviewer #2: Yes

6. Review Comments to the Author

Reviewer #1: Reviewer is still not convinced with few results coming from the tables. Also in spite of correcting the references, there are still modifications to be done as in few references the authors have provided the year in brackets after the name of the author and in some places they have provided the year without brackets after the journal name. Not sure what style they have followed.

There are some other observations too

>“Our study contradicts the existing literature and shows that the odds of older adults suffering from diarrhoea were higher among those who belonged to richer section of population [44, 53].

the results are opposite in table 3. Also no argument has been provided in the discussion part. While explaining the results of table 3, authors also have missed to write about this result.

Inspite of pointing this in the last comments this has not been mentioned in the results section. Authors should mention everything coming from their study. Presentation of only selective results from table 3 is not recommended.

>This is a very important finding.

Source of drinking water - adults suffering from diarrhea

Unimproved 10.9 %

Improved 15.2 %

This has yet not been written in the results section.

>The highest rural-urban difference in the prevalence of diarrhoea was observed among older adults who lived in kutcha houses. Studies conducted in Bangladesh and Ethiopia revealed the same findings [40–44].

References 40 to 44 includes india, Indonesia,along with Bangladesh and Ethiopia. They are not only on Bangladesh and Ethiopia. Also ref 40 talks about "Among the individual food-hygiene variables, children who lived in the house with less dirty sewage had significantly lower diarrhea prevalence" and not directly on kutcha pakka houses. Authors can be more descriptive while citing references so that the sentences becomes self explanatory.

>Also reference 43 by Luby did only talk about handwashing practices and diarrhoea among children. Why are authors citing references which are not talking about kutcha pakka houses. This is a wrong practice.

Also ref 44 is on Risk of Adverse Pregnancy Outcomes among Women Practicing Poor Sanitation in Rural India: A Population-Based Prospective Cohort Study.

>Findings from a previous study supported our results that older adults with high education had lower risk of diarrhoea [45]

Ref 45 is on Incidence and Correlates of Diarrhea, Fever, Malaria and Weight Loss Among Elderly and Non-Elderly

Internally Displaced Parents in Cibombo Cimuangi in the Eastern Kasai Province, Democratic Republic of the Congo. This talked about role of spouse's education. I am not sure how are the authors citing there references linking to their studies directly.

>Logistic regression results reveal that the prevalence of diarrhoea was positively associated with higher age of older adults, who belonged to Scheduled Tribe (22 per cent higher risk) and OBC social group (24 per cent higher risk). This finding is consistent with a study carried out in India [46].

Ref 46 is on under 5 children in India. Authors should not directly link to them. Even if linking they should mention about the study done among under 5 children. The representation not proper.

>A higher concentration of diarrhoea was found in central and northeastern parts than in southern states of India. This could be because of unequal access to health care facilities, use of untreated drinking and low hygiene practices.

it will be better if the authors can find any literature supporting their argument.

>Discussion part is still not adequate as this study has many important and striking findings.

>Moreover, the study reveals that older adults who belong to the Christian religion were more likely to have diarrhoeal risk than Hindu older adults. However, this finding is inconsistent with previous research in India [37,46].

In reference number 37 and 46- both the studies are on Children and also mentioned about Muslim children suffering more than Other religion. I am not sure whether authors can use these literature to show inconsistency, as their own results are concerned about the Christians and Hindus. and also on older adults.

So many mistakes and improper use of literature in Discussion part is unacceptable.

I suggest all the authors should go through the manuscript attentively and focus on their results and the discussion part. More literature review is required. They should also go through few other published papers and follow how to write the discussion part. They should resubmit the manuscript when they feel it is ready for publication.

Reviewer #2: If possible, please prepare the map of India propoerly by using ARC GIS software. Put lat-long and other spatial details in the map.

7. PLOS authors have the option to publish the peer review history of their article (what does this mean?). If published, this will include your full peer review and any attached files.

Reviewer #1: No

Reviewer #2: **Yes: **Tushar Dakua

---

## [Author Response · Author response to Decision Letter 2]

21 Feb 2022

Reviewer #1: Reviewer is still not convinced with few results coming from the tables. Also in spite of correcting the references, there are still modifications to be done as in few references the authors have provided the year in brackets after the name of the author and in some places they have provided the year without brackets after the journal name. Not sure what style they have followed.

Response: The authors have followed Vancouver style using Zotero. The references are now edited using the same software. 

There are some other observations too

“Our study contradicts the existing literature and shows that the odds of older adults suffering from diarrhoea were higher among those who belonged to richer section of population [44, 53].

the results are opposite in table 3. 

Also no argument has been provided in the discussion part. While explaining the results of table 3, authors also have missed to write about this result.

Inspite of pointing this in the last comments this has not been mentioned in the results section. Authors should mention everything coming from their study. Presentation of only selective results from table 3 is not recommended.

Response: Dear reviewer, relationship of diarrhoea with wealth quintile has been added in results section of table 3. Also we have added in the discussion section.

This is a very important finding. Source of drinking water - adults suffering from diarrhea.

Unimproved 10.9 % 

Improved 15.2 % 

This has yet not been written in the results section.

Response: This now added in the result section. 

>The highest rural-urban difference in the prevalence of diarrhoea was observed among older adults who lived in kutcha houses. Studies conducted in Bangladesh and Ethiopia revealed the same findings [40–44].

References 40 to 44 includes India, Indonesia, along with Bangladesh and Ethiopia. They are not only on Bangladesh and Ethiopia. 

Also ref 40 talks about "Among the individual food-hygiene variables, children who lived in the house with less dirty sewage had significantly lower diarrhea prevalence" and not directly on kutcha pakka houses. Authors can be more descriptive while citing references so that the sentences becomes self explanatory.

Response: Dear Reviewer, we apologize for the mistake. We have now added all the countries mentioned in reference 40-44. However, we have deleted India since it was on Risk of Adverse Pregnancy Outcomes among Women Practicing Poor Sanitation in Rural India: A Population-Based Prospective Cohort Study (Reference 44).

Dear Reviewer, we have rephrased the sentence.

>Also reference 43 by Luby did only talk about handwashing practices and diarrhoea among children. Why are authors citing references which are not talking about kutcha pakka houses. This is a wrong practice. Also ref 44 is on Risk of Adverse Pregnancy Outcomes among Women Practicing Poor Sanitation in Rural India: A Population-Based Prospective Cohort Study.

Response: Dear Reviewer, we apologize for the mistake. We have now specifically mentioned what each of the study deals with. Also, we have deleted reference 44 in this context.

>Findings from a previous study supported our results that older adults with high education had lower risk of diarrhoea [45]

Ref 45 is on Incidence and Correlates of Diarrhea, Fever, Malaria and Weight Loss Among Elderly and Non-Elderly

Internally Displaced Parents in Cibombo Cimuangi in the Eastern Kasai Province, Democratic Republic of the Congo. This talked about role of spouse's education. I am not sure how are the authors citing there references linking to their studies directly.

Response: Dear Reviewer, we have removed reference 45 and added “Incidence and Determinants of Acute Diarrhoea in Malaysia: A Population-based Study”.

>Logistic regression results reveal that the prevalence of diarrhoea was positively associated with higher age of older adults, who belonged to Scheduled Tribe (22 per cent higher risk) and OBC social group (24 per cent higher risk). This finding is consistent with a study carried out in India [46].

Ref 46 is on under 5 children in India. Authors should not directly link to them. Even if linking they should mention about the study done among under 5 children. The representation not proper.

Response: Dear Reviewer, we apologise for the mistake. We have made corrections accordingly and mentioned that the study was done among children under the age of five years.

>A higher concentration of diarrhoea was found in central and northeastern parts than in southern states of India. This could be because of unequal access to health care facilities, use of untreated drinking and low hygiene practices.

it will be better if the authors can find any literature supporting their argument.

Response: Dear Reviewer, though we were unable to support our study finding with literature on older adults, we could present some studies that dealt with children. 

>Discussion part is still not adequate as this study has many important and striking findings.

>Moreover, the study reveals that older adults who belong to the Christian religion were more likely to have diarrhoeal risk than Hindu older adults. However, this finding is inconsistent with previous research in India [37,46].

Response: Dear Reviewer, we have tried to improve the discussion section.

In reference number 37 and 46- both the studies are on Children and also mentioned about Muslim children suffering more than Other religion. I am not sure whether authors can use these literature to show inconsistency, as their own results are concerned about the Christians and Hindus. and also on older adults.

Response: Dear Reviewer, we agree with you. However, due to lack of literature on diarrhea among older adults in India we could not support our finding. Hence, relied on literature on children in India.

So many mistakes and improper use of literature in Discussion part is unacceptable.

I suggest all the authors should go through the manuscript attentively and focus on their results and the discussion part. More literature review is required. They should also go through few other published papers and follow how to write the discussion part. They should resubmit the manuscript when they feel it is ready for publication.

Response: Dear Reviewer, we have tried to improve the discussion section.

Reviewer #2: If possible, please prepare the map of India propoerly by using ARC GIS software. Put lat-long and other spatial details in the map.

Response: Dear reviewer, the map is made using Arc GIS software. The spatial details are added.

---

## [Editor Report · Decision Letter 3]

23 Feb 2022

Rural-urban differentials in the prevalence of diarrhoea among older adults in India: Evidence from Longitudinal Ageing Study in India, 2017-18

PONE-D-21-09569R3

Dear Kumar,

We’re pleased to inform you that your manuscript has been judged scientifically suitable for publication and will be formally accepted for publication once it meets all outstanding technical requirements.

Kind regards,

Shah Md Atiqul Haq

Section Editor

PLOS ONE

Additional Editor Comments (optional):

Dear authors,

Your paper is now accepted.
---

## [Editor Report · Acceptance letter]

3 Mar 2022

PONE-D-21-09569R3 

Rural-urban differentials in the prevalence of diarrhoea among older adults in India: Evidence from Longitudinal Ageing Study in India, 2017-18 

Dear Dr. Kumar:

I'm pleased to inform you that your manuscript has been deemed suitable for publication in PLOS ONE. Congratulations! Your manuscript is now with our production department. 

Kind regards, 

on behalf of

Dr. Shah Md Atiqul Haq 

Section Editor

PLOS ONE